# Concordance of *Helicobacter pylori* Detection Methods in Symptomatic Children and Adolescents

**DOI:** 10.3390/microorganisms13030583

**Published:** 2025-03-04

**Authors:** Camila Cabrera, Yanira Campusano, Joaquín Torres, Dinka Ivulic, Valeria Galvez, Diego Tapia, Vicente Rodríguez, Anne Lagomarcino, Alejandra Gallardo, Francisco Alliende, Marcela Toledo, Gabriela Román, Francisca Jaime, Mónica González, Pamela Marchant, Marianela Rojas, Juan Ignacio Juanet, Mónica Villanueva, Juan Cristobal Ossa, Felipe Del Canto, Tomeu Viver, Miguel O’Ryan, Yalda Lucero

**Affiliations:** 1Microbiology and Mycology Program, Institute of Biomedical Sciences, Faculty of Medicine, University of Chile, Santiago 8380453, Chile; camila.cabrera@ug.uchile.cl (C.C.); yaniracampusano@ug.uchile.cl (Y.C.); joaquintorres@ug.uchile.cl (J.T.); ivulic.dinka@gmail.com (D.I.); valeria.galvez.s@usach.cl (V.G.); diego.tapia.l@ug.uchile.cl (D.T.); vicente.rodriguez.s@ug.uchile.cl (V.R.); anne.lagomarcino@uchile.cl (A.L.); felipedelcanto@uchile.cl (F.D.C.); moryan@uchile.cl (M.O.); 2Pathology Department, Clínica Alemana de Santiago, Facultad de Medicina, Clínica Alemana-Universidad del Desarrollo, Santiago 7650568, Chile; agallardos@alemana.cl; 3Pediatric Gastroenterology Unit, Clínica Alemana de Santiago, Facultad de Medicina, Clínica Alemana-Universidad del Desarrollo, Santiago 7650568, Chile; falliende@alemana.cl (F.A.); mjaime@alemana.cl (F.J.); pmarchant@alemana.cl (P.M.); mvillanuevac@alemana.cl (M.V.); 4Pediatric Gastroenterology Unit, Hospital Roberto del Río, Santiago 8380000, Chile; marcelatoledo5@gmail.com (M.T.); monigonzaya@gmail.com (M.G.); rojasv.marianela@gmail.com (M.R.); jijuanet@gmail.com (J.I.J.); 5Pediatric Gastroenterology Unit, Hospital Exequiel Gonzalez Cortés, Santiago 8900085, Chile; g.romanmatamala@gmail.com; 6Pediatric Gastroenterology Unit, Hospital Padre Hurtado, Santiago 8880465, Chile; 7Pediatric Gastroenterology Unit, Hospital Luis Calvo Mackenna, Santiago 7500539, Chile; jcossa@gmail.com; 8Marine Microbiology Group, Department of Animal and Microbial Biodiversity, Mediterranean Institute for Advanced Studies (IMEDEA, CSIC-UIB), 07190 Esporles, Spain; tviver@imedea.uib-csic.es; 9Instituto de Sistemas Complejos de Ingeniería (ISCI), Santiago 8370398, Chile; 10Department of Pediatrics and Pediatric Surgery (Northern Campus), Hospital de Niños Roberto del Río, Faculty of Medicine, University of Chile, Santiago 8380000, Chile

**Keywords:** *Helicobacter pylori*, children, adolescents, histology, RT-PCR, rapid urease test, serology, gastric microbiome

## Abstract

Background: *Helicobacter pylori* is the most prevalent chronic bacterial infection globally, acquired mostly during childhood. It is associated with chronic gastritis, peptic ulcer disease, and gastric cancer. Due to challenges in culturing *H. pylori*, diagnostic reference standards often rely on combining ≥2 non-culture, biopsy-based methods. Histology with Giemsa staining is widely used in clinical settings due to its low cost and reliable performance. Methods: This study evaluated the concordance between histology with Giemsa staining as the reference standard and other diagnostic methods, including the rapid urease test (RUT), ureA RT-PCR, 16S sequencing, and anti-*H. pylori* serum IgG. Positive percent of agreement (PPA), negative percent of agreement (NPA) and concordance kappa index were calculated. Results: A total of 120 patients (41 positive and 79 negative by Giemsa staining) were analyzed. Among the methods tested, RT-PCR for *ureA* showed the best performance (PPA = 94.7%, NPA = 98.6%, kappa = 0.939), while RUT underperformed compared with expectations (PPA = 65.9%, NPA = 97.5%, kappa = 0.681). Serology had the lowest performance (PPA = 53.7%, NPA = 96.1%, kappa = 0.548). Conclusions: The combination of histology with Giemsa staining and *ureA* RT-PCR achieved the highest detection rate and strongest agreement.

## 1. Introduction

*Helicobacter pylori* is a microaerophilic, Gram-negative bacterium recognized as the primary cause of chronic gastritis, peptic ulcer disease (PUD), gastric MALT lymphoma, and gastric adenocarcinoma [1,2]. Its global prevalence is estimated to be approximately 50% in adults and 20% to 50% in children, depending on the population studied [3]. The infection is more common in regions with suboptimal socio-sanitary conditions, including certain areas in Latin America [4]. In these areas, it is mainly acquired during childhood through direct contact with infected individuals, often persisting for decades in the absence of eradication therapy [2,5].

Diagnosis of *H. pylori* infection can be established using either invasive or non-invasive methods. Invasive techniques, which involve analyzing gastric biopsies obtained during upper endoscopy, allow for direct detection of the bacterium and associated mucosal lesions [5]. This approach is particularly recommended in pediatric patients, as eradication therapy is advised only in the presence of mucosal damage [5].

Biopsy samples can be evaluated using culture, rapid urease tests (RUT), histology, or molecular biology techniques. Although culture is considered the gold standard for *H. pylori* diagnosis, its routine implementation is challenging due to the bacterium’s demanding growth requirements. The need for specific microaerophilic conditions, selective media, and prolonged incubation makes cultivation difficult in standard clinical laboratories, resulting in low recovery rates and limiting its practical utility [4]. Consequently, international consensus guidelines recommend combining ≥2 biopsy-based non-culture tests to confirm infection [5]. However, discrepancies between these methods are common in practice, and diagnostic accuracy may be lower than reported in clinical trials due to technical factors, such as biopsy size, bacterial density in the samples, and processing variability [5,6,7]. Patient-related factors may also decrease test performance, including the current or recent use of proton pump inhibitors (PPI) or antibiotics, mucosal bleeding, or the presence of intestinal metaplasia [6].

Non-invasive methods for *H. pylori* detection include the urea breath test (UBT), stool antigen testing, and serology for anti-*H. pylori* IgG. While serology is recognized as a sensitive, cost-effective, and easily implemented method, its diagnostic accuracy in detecting active infection is limited, as antibodies can remain detectable long after eradication, potentially leading to false positive results [4,5,8].

This study aims to evaluate the concordance among various *H. pylori* detection methods in symptomatic Chilean children undergoing endoscopy, using histopathology with Giemsa staining as the reference standard. Given the limited availability of culture in real-world clinical settings in Latin America, we selected Giemsa-stained histology as a pragmatic and widely accessible comparator. In the absence of a true gold standard for comparison, we assessed diagnostic concordance by calculating the Positive Percent Agreement (PPA) and Negative Percent Agreement (NPA), following CLSI and FDA guidelines [9,10,11]. The evaluated methods included ureA RT-PCR, rapid urease test (RUT), and 16S sequencing from biopsy samples, as well as anti-*H. pylori* IgG detection in blood.

## 2. Materials and Methods

### 2.1. Study Design

This prospective, multicenter, observational, and analytical study was conducted between 2019 and 2023 at 5 tertiary Centers (Roberto del Río Hospital, Exequiel González Cortés Hospital, Luis Calvo Mackenna Hospital, Padre Hurtado Hospital, and Clínica Alemana de Santiago). Consecutive patients aged 8 to 20 years undergoing upper gastrointestinal endoscopy (UGIE) for dyspepsia or abdominal pain were invited to participate. Exclusion criteria included continuous use of proton pump inhibitors or ranitidine within two weeks before endoscopy, antibiotic use within four weeks prior, active upper gastrointestinal bleeding, or coagulopathy. Blood samples and endoscopic gastric biopsies were collected after obtaining informed consent and assent when applicable. The Institutional Review Boards (IRBs) of the participating hospitals and the Ethics Committee of the Faculty of Medicine, University of Chile, approved the study protocol.

### 2.2. Sample Collection

During the UGIE, five gastric biopsies were obtained: three from the antrum, angle, and gastric body for histological analysis; one from the angle for a RUT and 16S rRNA sequence analysis; and one from the antrum for detection of *ureA* gene transcripts using reverse transcription-polymerase chain reaction (RT-qPCR). Additionally, 10 mL of blood was collected in EDTA tubes to determine anti-*H. pylori* IgG antibodies (Figure 1).

### 2.3. Histological Analysis

Since histology with Giemsa staining of gastric biopsies is widely used in clinical practice due to its low cost and reliable performance, we compared the concordance of *H. pylori* detection by the other methods with this technique as the reference. The samples were transported in formalin to the Pathology Department of Clínica Alemana de Santiago. After paraffin fixation, the samples were stained with Giemsa and examined by a pathologist blinded to the patient’s *H. pylori* status. The presence of *H. pylori*-like bacilli in the superficial gastric mucosa was a positive result.

### 2.4. Rapid Urease Test

The Helicotec^®^ UT Plus (Strong Biotech Corporation™, Taipei, Taiwan) was used to detect *H. pylori* urease activity in gastric biopsies collected during UGIE. A positive result was defined as a color change from yellow to pink after 1 h of incubation at room temperature. Following the test, the biopsy specimen was transported at −20 °C and subsequently stored at −80 °C until DNA extraction.

### 2.5. 16S rRNA Sequence Analysis

DNA extraction was performed using the QIAmp^®^ DNA Micro Kit (QIAGEN, Germantown, MD, USA). The quality of the extracted DNA was assessed by electrophoresis, and its concentration was measured using the Synergy™ HT system. DNA samples were stored at −80 °C until shipment to MR DNA (Shallowater, TX, USA), for amplification and sequencing of the V3 and V4 variable regions of the *16S* rRNA gene using the Illumina MiSeq platform.

The amplicon sequence variant (ASV) analysis was performed using the Qiime2 (version 2020.2.0) bioinformatic platform [12]. Raw sequencing data were trimmed using the following parameters: --p-trunc-len-f 280, --p-trunc-len-r 220, --p-trim-left-f 19 and --p-trim-left-r 22. Amplicon sequence variant analyses (ASV) were obtained with DADA2 software (version 1.10.1) implemented in Qiime2 (2020.2.0). The representative sequence for each ASV was aligned using the non-redundant SILVA REF 138.1 database and the SINA tool (version v1.3.1) implemented in the ARB program [13,14]. The aligned sequences were then inserted in the SILVA REF 138.1 pre-existing tree using the parsimony tool available in the ARB package (version v6.0.6). Finally, the phylogenetic tree was manually supervised, and the single isolated phylogenetic sub-branches containing the query sequences and at least one representative sequence were grouped into OPUs (operational phylogenetic units) [15,16]. Here, we only show the *H. pylori*-associated OPU detection results to compare its yield with the other methods. Microbiota diversity and composition analysis is beyond the scope of this study and will be included in another manuscript that is being prepared.

### 2.6. ureA Expression Detection

RNA was extracted from an antral biopsy stored in RNA Save^®^ solution using the QIAGEN™ RNeasy^®^ Micro Kit (QIAGEN, Germantown, MD, USA). Electrophoresis assessed the RNA’s quality, and its quantity was measured using the Synergy™ HT system. The RNA was then reverse-transcribed into complementary DNA (cDNA) using the Thermo Scientific™ RevertAid RT^®^ Kit (Waltham, MA, USA) with random primers and stored at −20 °C until further processing.

The *ureA* gene was subsequently amplified using the Magnetic Induction Cycler qPCR (Bio Molecular Systems, Upper Coomera, QLD, Australia) with the SensiFAST SYBR No-ROX kit (Meridian, London, UK). The reaction mixture included 5 µL of master mix, 2 µL of water, 400 nM of each primer (F-CGTGGCAAGCATGATCCAT and R-GGGTATGCACGGTTACGAGTTT), and 2 µL of cDNA diluted 1:10 in water, resulting in a final reaction volume of 10 µL. Amplification conditions included an annealing temperature of 60 °C [17]. The *H. pylori* ATCC 43504 strain was used as a positive control, and nuclease-free water served as a negative control. Amplification was deemed positive if a detectable curve was observed at any Ct value with a melting temperature of 82 °C.

### 2.7. Antibody Detection

The blood samples were transported at 4 °C and processed within four hours. After centrifugation at 200× *g* for 10 min, the plasma was separated and centrifuged again at 1000× *g* for an additional 10 min. The resulting supernatant was aliquoted and stored at −80 °C. IgG anti-*H. pylori* levels were measured by ELISA using the Gastropanel^®^ kit (GEBRAX, Freudenberg, Germany). A value of ≥30 EIU was considered positive. 

### 2.8. Data Management

Sample registration and storage were managed using NorayBanks^®^ (NorayBio, Derio, Spain) in compliance with Biobank standards. Demographic, clinical, and endoscopic data and laboratory procedure results were collected and managed using REDCap^®^ (Vanderbilt University, Nashville, TN, USA) electronic data capture tools hosted at the Faculty of Medicine of the University of Chile. Results for *H. pylori* detection from the RUT, microbiome analysis by *16S* rRNA sequencing, *ureA* gene expression, and antibody detection were compared to histology results, which served as the reference method.

### 2.9. Statistical Analysis

Statistical analysis was conducted using the stats v4.2.3 package in R (R Core Team, 2023, Vienna, Austria). Categorical variables were summarized as frequencies and percentages, while continuous variables were reported using central tendency and dispersion measures. The Chi-square or Fisher’s exact test was applied to compare categorical variables, depending on the number of observations per category. The Mann–Whitney U test was used to compare the only continuous variable (age), as its distribution was non-normal. To assess agreement between the reference method and the other diagnostic tests, the kappa coefficient, Positive Percent Agreement (PPA), and Negative Percent Agreement (NPA) were calculated.

## 3. Results

### 3.1. Patient Description

A total of 120 patients were included in the study, of whom 41 tested positive for *H. pylori* by histology with Giemsa staining. The demographic, clinical, endoscopic, and histological characteristics of the patients are summarized in Table 1. No significant differences were observed in the demographic and clinical features comparing *H. pylori*-positive (Hp (+)) and -negative (Hp (−)) patients. However, Hp (+) patients exhibited a significantly higher frequency of nodular gastropathy (*p* < 0.001) and demonstrated greater severity of histological gastritis (*p* < 0.001).

### 3.2. Detection of H. pylori by Different Methods and Concordance with Giemsa Staining

Samples from all 120 patients were analyzed using histology with Giemsa staining and RUT. One hundred and eighteen gastric samples were available for microbiome by *16S* sequence analysis, and the same number of plasma samples were available for anti-*H. pylori* IgG detection by ELISA, while 109 samples were available for the detection of *ureA* transcripts by RT-qPCR. Histology with Giemsa staining was the only semi-quantitative method, whereas all the others were qualitative.

Figure 2 shows the positivity rate of each test by patient, and Figure 3 illustrates the proportion of agreement between methods. Among the diagnostic techniques, *ureA* RT-PCR demonstrated the highest concordance with histology (PPA 94.7%, NPA 98.6%, Kappa index 0.9). This was followed by *16S sRNA* sequencing (PPA 82.9%, NPA 96.1%, Kappa index 0.8). In contrast, the RUT demonstrated a lower detection yield than previously reported and anti-*H. pylori* IgG exhibited the lowest agreement rate with histology.

## 4. Discussion

*H. pylori* infection remains a major global health concern, particularly in Latin America, where its high prevalence contributes significantly to the burden of gastric diseases. Because culture-based methods, the gold standard for diagnosis, are not widely available or feasible in routine clinical practice [18], a combination of alternative techniques is often employed to detect the infection. Each diagnostic method has strengths and limitations, and no universally recommended combination exists. This study compared detection rates by multiple diagnostic techniques for *H. pylori*, using histology with Giemsa staining as the reference standard due to its availability and reliability.

Due to the absence of culture as the gold standard technique in this study and generally in procedures performed in Latin America, it is not possible to calculate sensitivity, specificity, positive predictive value, and negative predictive value. However, according to the CLSI EP12-A3 guideline, it is possible to calculate the positive and negative agreement percentages for diagnostic method comparisons. These parameters allow us to analyze and compare methods using a reference technique different from the gold standard, as in the case of this study [9].

Histology offers variable sensitivity (69–93%) and specificity (87–90%), which can be improved with techniques such as modified Giemsa, immunohistochemistry, and FISH [19,20]. Its advantages include a lower cost than other diagnostic methods, such as molecular biology, adding the capacity to evaluate mucosal inflammation alongside the diagnosis of infection. However, histology is operator-dependent and may fail to detect Gram-negative bacilli in cases of low bacterial load or gastric mucosal atrophy, potentially reducing sensitivity [20].

In this study, histology with Giemsa staining detected *H. pylori* in 41 of 120 patients (34.2%), consistent with the expected prevalence in this population [21]. Demographic and clinical characteristics did not differ significantly between Hp (+) and Hp (−) patients, corroborating findings that symptoms of dyspepsia are not specific to *H. pylori* infection [22]. However, Hp (+) patients exhibited a significantly higher frequency of nodular gastropathy and more severe histological gastritis, reinforcing *H. pylori*’s role in gastric pathology, particularly the development of chronic gastritis and other gastropathies, emphasizing that *H. pylori* is not an innocent bystander in gastric disease [23].

Regarding the other diagnostic methods, the *ureA* RT-PCR demonstrated the highest agreement with histology, indicating that molecular biology is a highly reliable approach for detecting *H. pylori* infection and may be the preferred option to complement histology for confirmation when available. We observed a trend toward an inverse relationship between the *ureA* Ct value in RT-qPCR and the relative abundance of *H. pylori* in histology. Although this association was not statistically significant, it suggests that RT-qPCR could provide semi-quantitative information, similar to histology, and may be useful as a complementary tool for clinical decision-making. Furthermore, this method is not dependent on the operator’s experience and is more reproducible and consistent than histology. One sample was positive by *ureA* RT-PCR, 16S sequence analysis, and RUT but negative by Giemsa staining, suggesting a possible false-negative of histology result and higher sensitivity of molecular methods. Conversely, two samples were positive by histology but negative by *ureA* RT-PCR and other methods, which may indicate a false-positive histology result due to the presence of *H. pylori*-like bacilli, that could belong to a different bacterial species, such as *H. heilmannii*, *H. bizzozeroni*, or *Pseudomonas fluorescence* [24]. However, these species could not be identified in the *16S* analysis of these two samples.

On the other hand, the 16S rRNA gene sequencing method showed excellent NPA (96.1%), but slightly lower PPA compared to histology. This discrepancy may result from the broad scope of microbiome analysis, which detects a range of bacteria, including those unrelated to *H. pylori*. Nevertheless, *16S* rRNA sequencing provides valuable insights into the gastric microbiome and could be helpful for broader studies of *H. pylori*’s ecological niche and its interactions with other microbial species. Finally, the accuracy of species classification using sequence analysis of the V3 and V4 regions of the *16S* rRNA gene is lower compared to full gene or metagenomic analysis [25,26], suggesting that further studies are needed to compare these methods, considering not only their high performance but also the higher costs in the context of resources constraints.

The RUT, a commonly used test in clinical practice, showed a lower detection yield than what has been described in other studies reporting sensitivity >90% and specificity near 95% [27,28]. This finding aligns with more recent studies reporting reduced sensitivity of RUT and lower detection rates compared to histology and UBT, particularly in populations with lower bacterial loads [29,30], which may apply to the patients in this series. To achieve truly positive RUT results, a bacterial load of at least 10⁵ organisms in the biopsy sample is required [31]. Any factor that reduces this bacterial concentration can affect the test’s sensitivity. In pediatric patients, infections with bacterial loads below this threshold have been described, where even the time required to obtain a positive result may be influenced by this variable. However, three patients described as having an “abundant quantity of bacilli” and two patients with a “moderate quantity of bacilli”, who also tested positive by molecular biology, had negative RUT results. This suggests that other factors may contribute to the negative RUT findings. Additionally, multiple studies have highlighted the importance of the number of biopsies in RUT performance [32]. Increasing the number of samples processed by RUT enhances the test’s efficiency and shortens the time needed to obtain a positive result. Seo et al. reported increased RUT positivity when using three biopsy samples instead of one [33]. Similarly, Siddique et al. demonstrated a higher positivity rate with more samples [32]. However, in the present study, ethical considerations limited RUT samples to a single biopsy, which may have impacted its performance. Furthermore, the use of antibiotics, proton pump inhibitors, or bismuth can reduce bacterial load, potentially affecting test sensitivity. Nonetheless, following international guidelines, we excluded patients who had received these treatments within the two and four weeks prior to testing, respectively. Therefore, we can confidently rule out these common causes of false negatives as contributors to the observed reduction in RUT performance.

The sensitivity of the RUT test is also reduced in cases of gastric atrophy and intestinal metaplasia [29]. In the Hp (−) group, we identified one patient with mucosal gastric atrophy and three patients with intestinal metaplasia, but none of these patients were positive for *H. pylori* by any other method, suggesting that these were true negatives. On the other hand, we identified two patients with positive RUT, but negative histology. *H. pylori* is not the only urease-producing bacterium that can be detected in gastric mucosa, as microorganisms such as *Streptococcus* sp., *Staphylococcus* sp., *Gardnerella* sp., *Lactococcus* sp., *Enterococcus* sp., or *Helicobacter heilmannii* can lead to false positives [6,26]. This could be the case for one patient with positive RUT but negative by all other methods.

Anti-*H. pylori* IgG serology showed the lowest agreement with histology. While suitable for population studies due to its non-invasive nature and low cost, serology’s inability to distinguish between current and past infections limits its clinical utility. Its poor performance in this study discourages its use for diagnosing active infections [19,34,35].

One limitation of this study is that we relied on histology as the reference standard, which, despite its high specificity, has limitations in sensitivity, particularly in cases of low bacterial load or uneven distribution of *H. pylori* in the gastric mucosa. However, this choice reflects methodological accessibility in Latin America, ensuring the relevance of our findings to similar settings. Performance was further controlled by using a single, experienced pathologist blinded to the patient’s *H. pylori* status. It is also important to note that in our series, there were no patients with peptic ulcer disease, likely due to our stringent exclusion criteria, which prevented the recruitment of individuals recently treated with proton pump inhibitors or those showing evidence of active upper-gastrointestinal bleeding.

The Urea Breath Test (UBT) and stool antigen testing (SAT) were excluded from this analysis because we intended to compare tests used for initial diagnosis in symptomatic patients, and these two tests are used primarily during follow-up to assess eradication success after treatment.

In conclusion, our findings underscore the importance of using reliable and sensitive diagnostic methods to detect *H. pylori* infection. The *ureA* RT-qPCR method demonstrated the highest concordance and accuracy compared to histology. It should be considered a more robust complement to histology than other methods in clinical and research settings. While *16S rRNA* sequencing offers valuable microbiome insights, it is better suited for research contexts due to its complexity and cost. The lower sensitivity of the RUT and serological tests highlights the need for careful interpretation and use of more accurate detection methods, especially in high-prevalence regions like Latin America.

Future studies should focus on more extensive and diverse populations to validate these findings and explore cost-effective diagnostic strategies suitable for different clinical settings.

## Figures and Tables

**Figure 1 microorganisms-13-00583-f001:**
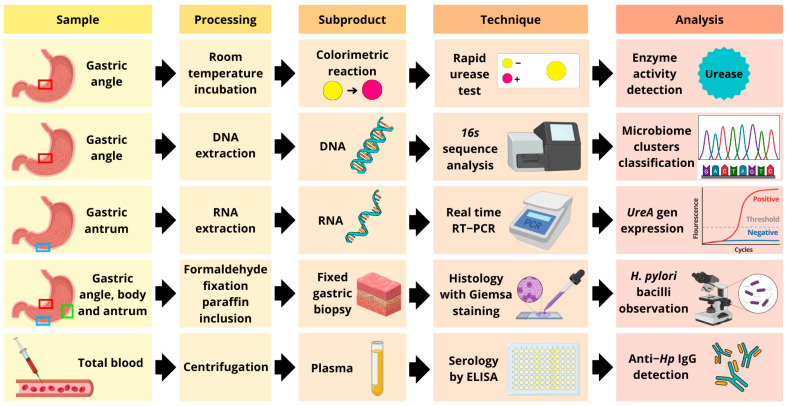
Flow chart of sample processing.

**Figure 2 microorganisms-13-00583-f002:**
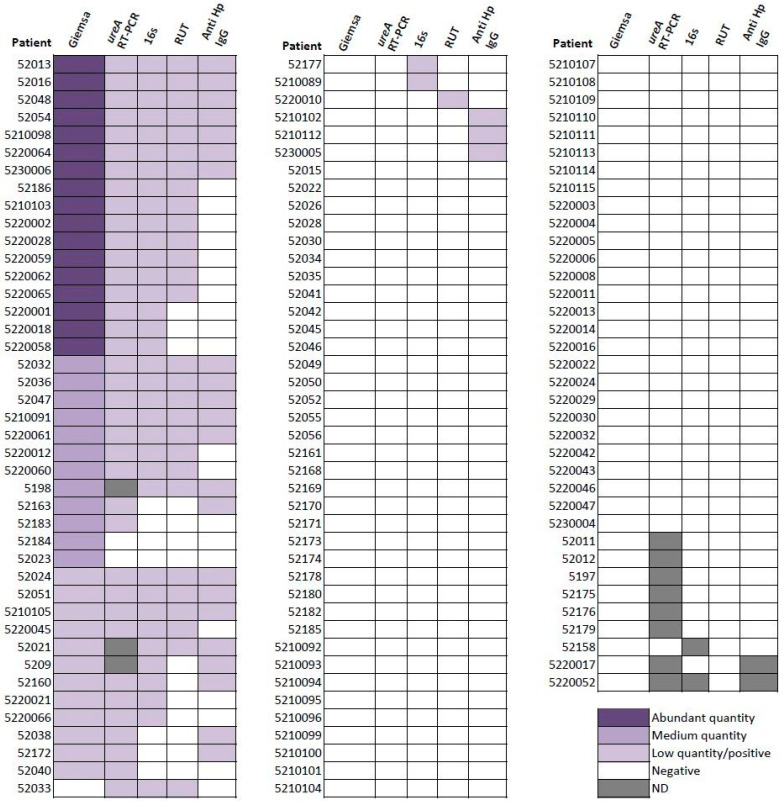
Results for *H. pylori* detection for each method by patient. All the positive samples for *H. pylori* are in purple. ND: Not determined.

**Figure 3 microorganisms-13-00583-f003:**
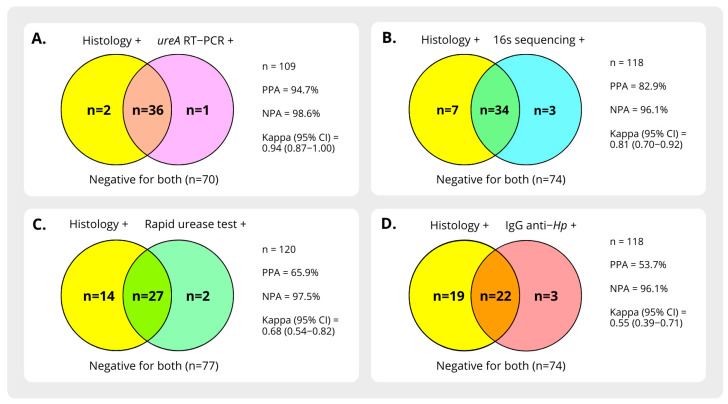
Venn diagram illustrating the positive agreement between histology with Giemsa staining of gastric mucosa samples and other diagnostic methods. (**A**) Comparison of gastric histology with ureA RNA detection by RT-qPCR in gastric samples. (**B**) Comparison of gastric histology with H. pylori detection by 16S rRNA amplification and sequence analysis. (**C**) Comparison of gastric histology with urease activity detection using the Rapid Urease Test (RUT). (**D**) Comparison of gastric histology with IgG anti-H. pylori detection in serum by ELISA.

**Table 1 microorganisms-13-00583-t001:** Demographic and clinical characteristics of the patients according to *H. pylori* detection by histology with Giemsa staining.

		Hp (+) N = 41	Hp (−) N = 79	Overall N = 120	OR	*p*
Demographics	Female, n (%)	24 (58.5)	59 (74.7)	83 (69.2)	0.482 (0.21–1.08)	0.069
	Age in years, median (interquartile range)	14 (8–20)	14 (8–20)	14 (8–20)		NS
Symptoms prompting endoscopy, n (%)	Epigastric pain	34 (82.9)	54 (68.4)	88 (73.3)	2.070 (0.82–5.77)	0.087
	Nocturnal abdominal pain	21 (51.2)	35 (44.3)	56 (46.6)	1.084 (0.50–2.33)	0.471
	Melena	3 (7.3)	10 (12.7)	13 (10.8)	0.549 (0.11–1.96)	0.539
	Persistent vomiting	10 (24.4)	14 (17.7)	24 (20.0)	1.550 (0.59–3.92)	0.386
	Hematemesis	2 (4.9)	5 (6.3)	7 (5.8)	1.491 (0.26–7.49)	1.000
	Weight loss	12 (29.3)	31 (39.2)	43 (35.8)	0.628 (0.26–1.40)	0.280
	Loss of apetite	24 (58.5)	37 (46.8)	61 (50.8)	1.517 (0.70–3.31)	0.224
	Anemia	4 (9.8)	10 (12.7)	14 (11.7)	0.512 (0.10–1.79)	NS
Family history, n (%)	Gastric cancer	12 (29.3)	12 (15.2)	24 (20.0)	2.005 (0.77–5.14)	0.067
	Duodenal Ulcers	8 (19.5)	19 (24.1)	27 (22.5)	0.684 (0.24–1.76)	0.572
	Gastritis	25 (61.0)	46 (58.2)	71 (59.2)	1.265 (0.58–2.81)	0.771
Gastric endoscopic finding, n (%)	Normal	4 (9.8)	43 (54.4)	47 (39.2)	10.52 (3.75–38.68)	<0.00001
	Nodular gastropathy	20 (48.8)	3 (3.8)	23 (19.2)		
	Congestive/erosive gastropathy	17 (41.5)	33 (41.8)	50 (41.7)		
Gastric histological findings, n (%)	No inflammation	1 (2.4)	22 (27.8)	23 (19.2)	13.45 (2.65–329.37)	0.0004
	Mild inflammation	7 (17.1)	55 (69.6)	62 (51.7)		
	Moderate inflammation	31 (75.6)	2 (2.5)	33 (27.5)		
	Severe inflammation	2 (4.9)	0 (0.0)	2 (1.7)		

## Data Availability

Raw 16S rRNA gene sequences are available upon reasonable request from the corresponding author. The data are not publicly available at this time as they are part of an ongoing study and are intended for use in a future publication. After completing this work, the raw sequences will be accessible in a publicly available repository. All data are contained within the article.

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
