# Peer review of "Concordance of Helicobacter pylori Detection Methods in Symptomatic Children and Adolescents"

_microorganisms, 2025, doi:10.3390/microorganisms13030583_

Round 1
Reviewer 1 Report
Comments and Suggestions for Authors
I have reviewed the manuscript “Concordance of Helicobacter pylori detection methods in symptomatic children and adolescents” by Camila Cabrera and coauthors conducted a comparative study between different methods of diagnosis and detection of Helicobacter pylori in 120 Chilean patients. The results showed that the RT-PCR method was the most concordant with those revealed by histology and is suggested as a combination to obtain the best positivity rates. The study is basic; however, its information may be relevant to countries or regions with similar diagnostic challenges. I leave some suggestions that I believe can help improve the manuscript.
Abstract:
• Line 30: Check the positive and negative values, as they do not agree with the results shown in the body of the manuscript.
• .
Introduction
• Lines 65-66: Also add the main factors associated with false negative results.
Methods
• Line 87: Remove "expression" as it is an indirect measurement, it detects only RNA. Modify throughout the manuscript.
• Lines 133-134: The primers were designed in this study? If not, the authors must cite the reference.
• Line 136: Specify that the ATCC 43504 strain corresponds to a H. Pylori.
Results
• Lines 168-170: Why were the 120 samples not used for each analysis? Inform.
• Lines 170-171: If the authors use a real-time PCR method to detect HP, they obtain a ct value that can be used semiquantitative.
• Detection of H. pylori by Different Methods and Concordance with Giemsa Staining: I do not see the point of using gastric microbiome analysis with 16S if HP 16S amplification and sequencing could be performed directly. In any case, the microbiome analysis offers many more results that the authors should include, at least as supplementary material. From these results, more discussions can also be derived such as % HP between the microbiome, other coinfecting species, etc.
Discussion:
• Lines 211-214: There was some relationship between abundant quantity with Giemsa and the ct values of the RT-PCR. This could be discussed.
• Lines 221-222: Were any of these other Helycobater species detected by 16S? I reiterate the importance of showing the complete results of this analysis to better support discussing these results.
• Lines 224-227: I suggest removing these sentences since the molecular methods used in the present study do not allow us to obtain this information.
• Line 252: add the reference used.
Author Response
Santiago, February 7th 2025.
Dear Reviewer 1,
We are writing to respectfully request you to re-consider our manuscript entitled "Concordance of Helicobacter pylori detection methods in symptomatic children and adolescents" for publication in this Special Issue of Microorganisms Journal. We truly thank you for your thorough review of our manuscript which will certainly be enriched by the proposed comments.
Here we are sending a point-by-point answer for your comments:
Can be improved: Introduction, Research design, Presentation of results.
1) Page 1 line 30: Check the positive and negative values, as they do not agree with the results shown in the body of the manuscript.
Response: We thank you for your comment and we corrected the numbers.
2) Introduction: Lines 65-66: Also add the main factors associated with false negative results.
Response: We added the factors associated with false negative results for RUT as suggested between lines 61 and 64.
3) Methods: Line 87: Remove "expression" as it is an indirect measurement, it detects only RNA. Modify throughout the manuscript.
Response: We changed the term “expression” by “transcripts” in order to clarify as suggested.
4) Methods: Lines 133-134: The primers were designed in this study? If not, the authors must cite the reference.
Response: We included the original reference from where we extracted the primers.
5) Methods: Line 136: Specify that the ATCC 43504 strain corresponds to a H. Pylori.
Response: We added this information as requested.
6) Results: Lines 168-170: Why were the 120 samples not used for each analysis? Inform.
Response: We thank you for your inquiry. Some patients did not provide all the requested samples. We added an explanation in the manuscript.
7) Results: Lines 170-171: If the authors use a real-time PCR method to detect HP, they obtain a ct value that can be used semiquantitative.
Response: We agree with the comment, but we were interested only in detection, then level of expression was not calculated.
8) Results: Detection of H. pylori by Different Methods and Concordance with Giemsa Staining: I do not see the point of using gastric microbiome analysis with 16S if HP 16S amplification and sequencing could be performed directly. In any case, the microbiome analysis offers many more results that the authors should include, at least as supplementary material. From these results, more discussions can also be derived such as % HP between the microbiome, other coinfecting species, etc.
Response: We agree with the comment. However, the focus of this manuscript is only detection and comparison of different methods. Deep analysis of gastric microbiota goes beyond the aim of this paper and is included in other manuscript under preparation.
9) Discussion: Lines 211-214: There was some relationship between abundant quantity with Giemsa and the ct values of the RT-PCR. This could be discussed.
Response: We thank you for your comment. There was an inverse association between ureA Ct and relative abundance of H. pylori but this was not statistically significant. We added a comment in the discussion section.
10) Discussion: Lines 221-222: Were any of these other Helycobater species detected by 16S? I reiterate the importance of showing the complete results of this analysis to better support discussing these results.
Response: In these 2 patients no other Helicobacer species was detected by 16 sequence analysis. We included a comment in discussion.
11) Discussion: Lines 224-227: I suggest removing these sentences since the molecular methods used in the present study do not allow us to obtain this information.
Response: We thank you for the comment and removed the sentence.
12) Discussion: Line 252: add the reference used.
Response: We agree and added the reference.
We truly appreciate your consideration and hope you find our manuscript suitable for publication in this prestigious journal.
Sincerely yours,
Yalda Lucero, MD, PhD
Pediatric Gastroenterologist
Associate Professor
Faculty of Medicine
University of Chile
Reviewer 2 Report
Comments and Suggestions for Authors
In the present study Cabrera et al compared histology with Giemsa staining for H. pylori detection with other diagnostic methods, including two based on PCR, in pediatric population. Main comments:
1) Page 1 line 30: Hp+ and Hp- were 41/79, not 40/80.
2) In table 1, please report all p values, even if NS.
3) Why Authors did not calculate sensitivity, specificity and negative/positive predictive value?
4) Low performance of RUT should be better discussed (see Losurdo G et al, Minerva Gastroenterol 2023).
5) Paragraph 2.5: results of gastric microbiome analysis are missing.
Author Response
Santiago, February 7th 2025.
Dear Reviewer 2,
We are writing to respectfully request you to re-consider our manuscript entitled "Concordance of Helicobacter pylori detection methods in symptomatic children and adolescents" for publication in this Special Issue of Microorganisms Journal. We truly thank you for your thorough review of our manuscript which will certainly be enriched by the proposed comments.
Here we are sending a point-by-point answer for your comments:
Can be improved: Conclusions.
Must be improved: Research design, Methods description, Presentation of results.
In general, we improved the writing and description of the methodology and results section.
1) Page 1 line 30: Hp+ and Hp- were 41/79, not 40/80.
Response: We thank you for your comment, and we corrected the numbers.
2) In table 1, please report all p values, even if NS.
Response: All the p-values were reported in the new version of Table 1, as suggested, and an explanation of the calculations was added to the methods section.
3) Why Authors did not calculate sensitivity, specificity and negative/positive predictive value?
Response: We thank your comment and included an explanation in the discussion section as follows: “Due to the absence of culture as the gold standard technique in this study, and generally in Latin America, it is not possible to calculate sensitivity, specificity, positive predictive value, and negative predictive value. However, according to the CLSI EP12-A2 guideline, it is possible to calculate the positive and negative agreement percentages for diagnostic method comparisons.”
4) Low performance of RUT should be better discussed (see Losurdo G et al, Minerva Gastroenterol 2023).
Response: We appreciate your suggestion and have incorporated the following paragraph into the Discussion section: “To achieve truly positive RUT results, a bacterial load of at least 10⁵ organisms in the biopsy sample is required. Any factor that reduces this bacterial concentration can affect the test's sensitivity. In pediatric patients, infections with bacterial loads below this threshold have been described, where even the time required to obtain a positive result may be influenced by this variable. Additionally, multiple studies have highlighted the importance of the number of biopsies in RUT performance. A greater number of biopsy samples enhances the test’s efficiency and shortens the time needed to obtain a positive result. Seo et al. reported an increase in RUT positivity when using three biopsy samples instead of one. Similarly, Siddique et al. demonstrated a higher positivity rate with an increased number of samples. However, in the present study, ethical considerations limited RUT analysis to a single biopsy sample, which may have impacted its performance. Furthermore, the use of antibiotics, proton pump inhibitors, or bismuth can reduce bacterial load, potentially affecting test sensitivity. Nonetheless, in accordance with international guidelines, we excluded patients who had received these treatments within the two and four weeks prior to testing, respectively. Therefore, we can confidently rule out these common causes of false negatives as contributors to the observed reduction in RUT performance.”
5) Paragraph 2.5: results of gastric microbiome analysis are missing.
Response: We thank you for your observation. There was a marker on the paragraph that hid the text. We have already fixed that problem, and the text is now visible.
We appreciate your consideration and hope you find our manuscript suitable for publication in this prestigious journal.
Sincerely yours,
Yalda Lucero, MD, PhD
Pediatric Gastroenterologist
Associate Professor
Faculty of Medicine
University of Chile
Reviewer 3 Report
Comments and Suggestions for Authors
The manuscript compares current techniques used in human H. pylori detection. Four other techniques were compared to histology. While the manuscript was easy to read, there are a couple of minor suggestions:
1) The UBT has been used as a good indicator of Hp infection in other studies. However, in Line 262 the authors indicated that UBT was excluded. Have you considered that the limitations of UBT could be very similar to those of the RUT regarding sensitivity?
2) The main weakness was the “pre-selection” or inclusion of patients rather than excluding only those currently or previously undergoing antibiotic treatment and acid-pump inhibitors. I would prefer to move or provide a line sentence about this issue in the study's limitations as you have it now in lines 207-210.
Author Response
Santiago, February 7th 2025.
Dear Reviewer 3,
We are writing to respectfully request you to reconsider our manuscript entitled "Concordance of Helicobacter pylori detection methods in symptomatic children and adolescents" for publication in this Special Issue of Microorganisms Journal. We genuinely thank you for thoroughly reviewing our manuscript, which the proposed comments will enrich.
Here we are sending a point-by-point answer for your comments:
Can be improved: Study design, Conclusions.
In general, we improved the writing and description of the methodology and results section.
- The UBT has been used as a good indicator of Hp infection in other studies. However, in Line 262 the authors indicated that UBT was excluded. Have you considered that the limitations of UBT could be very similar to those of the RUT regarding sensitivity?
Response: We thank you for your comment and agree. Detection of H. pylori by UBT and RUT is based on the activity of urease expressed by strains living on gastric mucosa. Then, similar constraints may limit the performance of these methods, as the recent use of antibiotics, proton pump inhibitors, and bismuth.
- The main weakness was the “pre-selection” or inclusion of patients rather than excluding only those currently or previously undergoing antibiotic treatment and acid-pump inhibitors. I would prefer to move or provide a line sentence about this issue in the study's limitations as you have it now in lines 207-210.
Response: We thank you for this suggestion and move this sentence to the limitations section as suggested.
We truly appreciate your consideration and hope you find our manuscript suitable for publication in this prestigious journal.
Sincerely yours,
Yalda Lucero, MD, PhD
Pediatric Gastroenterologist
Associate Professor
Faculty of Medicine
University of Chile
Round 2
Reviewer 1 Report
Comments and Suggestions for Authors
The authors have satisfactorily responded or justified the observations made.
Author Response
Dear Reviewer, We thank all your valuable comments and suggestions, that improved the manuscript. Kind Regards, Yalda.Reviewer 2 Report
Comments and Suggestions for Authors
Pint 4: the suggested reference was not discussed.
All other answers were fine.
Author Response
Dear Reviewer
We thank your comments and added the reference you suggested.
Kind regards,
Yalda.